medicinal chemistry/organic chemistry

melanoma, thiazole, androstenone, flow cytometry, apoptosis, TUNEL assay

**Author for correspondence:**
Mohammad A. Alam
e-mail: malam@astate.edu

# Antimelanoma activities of chimeric thiazole–androstenone derivatives

Steven A. Chambers[1], Mathew Newman[1], Melissa M. Frangie[1], Alena V. Savenka[2], Alexei. G. Basnakian[2] and Mohammad A. Alam[1]

[1]Department of Chemistry and Physics, College of Science and Mathematics, Arkansas State University, Jonesboro, AR 72467, USA
[2]Department of Pharmacology and Toxicology, University of Arkansas for Medical Sciences, and Central Arkansas Veterans Healthcare System, Little Rock, AR 72205, USA

MAA, 0000-0003-0258-3732

The discovery of chimeric anti-melanoma agents is reported. These molecules are potent growth suppressors of melanoma cells *in vitro* with growth inhibition of 50% ($GI_{50}$) values as low as 1.32 µM. Compounds were more toxic to melanoma cells *in vitro* than commonly used anti-melanoma agent dacarbazine as measured by TUNEL assay. They induced both caspase-independent apoptosis evident by colocalization of TUNEL with endonuclease G (EndoG) and caspase-mediated apoptosis measured by colocalization of TUNEL with caspase-activated DNase (CAD). In addition, compounds **3** and **5** strongly induced oxidative injury to melanoma cells as measured by TUNEL colocalization with heme oxygenase-1 (HO1). Dacarbazine induced only caspase-independent apoptosis, which may explain why it is less cytotoxic to melanoma cells than compounds **3**, **4** and **5**.

This article has been edited by the Royal Society of Chemistry, including the commissioning, peer review process and editorial aspects up to the point of acceptance.

Malignant melanoma (MM), also known as cutaneous melanoma, is a cancer that originates from melanocytes [1]. Melanoma is the fifth and seventh most commonly diagnosed cancer in American men and women, respectively. Among American populations, the incidence and mortality rates for invasive melanoma are the highest in Caucasian Americans. The estimated new cases and deaths in 2019 were 96 480 and 7230, respectively, in the US alone [2]. MM is one of the most common cancers in young Americans, especially young women [3]. Metastasized MM is virtually incurable, and patients with advanced melanoma have median survival time of less than a year. Recently approved approaches, such as chemotherapy, biochemotherapy and targeted therapy have shown disappointing results [4]. Therefore, finding new anti-cancer drugs to cure this lethal malignancy is imperative.

**Figure 1.** Thiazole containing cancer drugs.

**Figure 2.** Thiazole containing androstenone derivatives as potent anti-melanoma agents.

Thiazole-derived compounds are well-known among drugs, natural products and pharmacologically active synthetic molecules [5]. Several thiazole derivatives are widely used to treat different types of cancers, and a number of these compounds are at different stages of drug development. Some of the approved drugs are dasatinib to treat Philadelphia chromosome-positive chronic myelogenous leukaemia (CML) and acute lymphoblastic leukaemia (ALL), dabrafenib to treat BRAF-positive cancers and ixabepilone aimed at breast cancer (figure 1) [6,7]. A number of natural products containing the thiazole moiety include: firefly luciferin, dolastatin E, mirabazole C and leinamycin [8]. These molecules show a wide range of biological activities, such as anti-cancer, antimicrobial, antimalarial, anti-tuberculosis and many other useful therapeutic properties [9]. A large number of synthetic derivatives have been reported for various pharmacological applications [9,10].

In our effort to develop new methodologies to synthesize bioactive compounds [11–14], we have synthesized a number of thiazole and thiazolo-androstenone derivatives [15–17]. We submitted 50 of the synthesized thiazole–androstenone derivatives to the National Cancer Institute (NCI) Development Therapeutics Program (DTP) to test against 60 cancer cell lines (https://dtp.cancer.gov/). These compounds were tested for cytotoxicity at a single dose of 10 µM. Several compounds showed potent activity against several cancer cell lines, and some of them were particularly potent against melanoma cell lines. Eight of these chimeric compounds (figure 2) were selected for further testing at five different concentrations to determine 50% growth inhibition ($GI_{50}$), total growth inhibition (TGI) and 50% lethal concentration ($LC_{50}$) values. Based on these data, we found a very promising structure–activity relationship of these molecules against melanoma cell lines. Overall, amino-thiazole derivatives (table 1, **1–7**) [15] were more toxic to melanoma cell lines than phenyl thiazole derivatives (**8**) [16]. Fluoro and oxy (hydroxy and methoxy) derivatives were the potent growth inhibitors of the tested cancer cell lines. LOX IMVI cell line was the most susceptible among all the melanoma cell lines used against the tested compounds (table 1). All eight compounds inhibited this cell line with

**Table 1.** Fifty per cent growth inhibition (GI$_{50}$) values of compounds **1–8** against the melanoma panel.

| melanoma cell line | GI$_{50}$ (µM) | | | | | | | |
|---|---|---|---|---|---|---|---|---|
| | **1** | **2** | **3** | **4** | **5** | **6** | **7** | **8** |
| LOX IMVI | 1.70 | 3.22 | 1.74 | 1.67 | 1.67 | 1.71 | 1.64 | 1.74 |
| MALME-3M | 2.18 | 21.2 | 3.45 | 1.49 | 1.94 | >100 | 1.99 | 1.80 |
| M14 | 3.63 | 7.88 | 3.29 | 2.97 | 2.36 | >100 | 1.85 | 6.62 |
| MDA-MB-435 | 3.01 | 3.49 | 2.72 | 2.53 | 2.74 | >100 | 3.02 | 2.97 |
| SK-MEL-2 | 17.0 | 17.1 | 3.00 | 2.86 | 1.86 | >100 | 2.19 | 2.74 |
| SK-MEL-28 | 3.37 | 20.5 | 1.88 | 1.32 | 1.52 | >100 | 2.31 | 1.53 |
| SK-MEL-5 | 4.57 | 10.2 | 2.62 | 5.70 | 2.02 | 1.88 | 1.86 | 4.05 |
| UACC-257 | 4.92 | 23.8 | 4.89 | 3.69 | 5.47 | >100 | 3.70 | 6.12 |
| UACC-62 | 2.67 | 9.54 | 3.03 | 2.08 | 1.54 | >100 | 1.72 | 1.90 |

**Table 2.** Total growth inhibition (TGI) values of potent compounds against the melanoma panel.

| cell line (melanoma) | TGI (µM) | | | | | | |
|---|---|---|---|---|---|---|---|
| | **1** | **2** | **3** | **4** | **5** | **7** | **8** |
| LOX IMVI | 3.17 | 10.8 | 3.18 | 3.15 | 3.21 | 3.13 | 3.36 |
| MALME-3M | 6.82 | >100 | 16.8 | 3.01 | 5.04 | 5.55 | 4.13 |
| M14 | >100 | 50.9 | 8.72 | >100 | 7.23 | 3.79 | >100 |
| MDA-MB-435 | >100 | >100 | 10.1 | >100 | 13.1 | >100 | >100 |
| SK-MEL-2 | >100 | 60.8 | 11.6 | 9.99 | 5.19 | 4.74 | 10.7 |
| SK-MEL-28 | 18.9 | >100 | 7.98 | 3.31 | 4.27 | 6.08 | 4.07 |
| SK-MEL-5 | 52.7 | 26.0 | 7.41 | >100 | 4.32 | 3.73 | 16.9 |
| UACC-257 | 79.8 | 94.5 | 28.8 | >100 | 64 | >10 | >100 |
| UACC-62 | 10.6 | 29.6 | 10.3 | 5.25 | 3.12 | 3.94 | 5.70 |

GI$_{50}$ values as low as 3.22 µM. Phenyl ethyl derivative (**1**) inhibited the growth of all the cell lines except SK-MEL-2. Phenyl substituted derivative (**2**) showed good activity against LOX IMVI and MDA-MB-435 cell lines with GI$_{50}$ values of 3.22 and 3.49 µM, respectively. This molecule (**2**) showed moderate growth inhibition against M14 and UACC-62 cell lines with GI$_{50}$ values less than 10 µM. Monosubstituted phenyl derivatives **3**, **4** and **5** showed very significant activities against all the tested cell lines with GI$_{50}$ values as low as 1.32 µM for SK-MEL-28. 2,4-Difluoro substituted phenyl derivative (**6**) is found to be a good growth inhibitor of LOX IMVI and SK-MEL-5 cell lines with GI$_{50}$ values of 1.71 and 1.88 µM, respectively, while it failed to show any significant activity against other cell lines. 2,4-Dimethyl substituted derivative (**7**) is a potent growth inhibitor of all the tested cell lines of the melanoma panel. Among all the reported thiobenzamide-derived thiazoles [16], only 3-fluorophenyl derivative (**8**) showed significant activity against the melanoma panel.

Compound (**2**) showed weak TGI values (greater than 48 µM) but compound (**6**) did not show any useful TGI values against the cell lines (electronic supplementary material, Information). *N*-Phenylethyl derivative (**1**) had the TGI values 3.17 and 6.82 µM against LOX IMVI and MALME-3M cell lines, respectively, but did not show any significant TGI against any other cell lines (table 2). Trifluoromethoxy (–OCF$_3$) substituted phenyl derivative (**3**) showed good TGI value against LOX IMVI cell line and moderate values against M14, SK-MEL-28 and SK-MEL-5 cell lines. A similar pattern was observed for other compounds (**4**, **5**, **7** and **8**).

These potent compounds also showed significant LC$_{50}$ values against LOX IMVI cell lines at approximately 6 µM concentration (table 3). Compound **5** showed good cytotoxicity for SK-MEL-5 and UACC-62 lines with LC$_{50}$ values 9.24 and 6.31 µM, respectively. A similar pattern of potency was

**Table 3.** $LC_{50}$ values of potent compounds against the melanoma panel.

| cell line (melanoma) | $LC_{50}$ (µM) | | | | | | |
|---|---|---|---|---|---|---|---|
| | 1 | 2 | 3 | 4 | 5 | 7 | 8 |
| LOX IMVI | 5.91 | 48.1 | 5.82 | 5.94 | 6.16 | 5.96 | 6.47 |
| MALME-3M | >100 | >100 | 84.5 | >100 | 26.7 | >100 | 9.48 |
| M14 | >100 | >100 | 46.6 | >100 | 81.4 | >100 | >100 |
| MDA-MB-435 | >100 | >100 | 44.2 | >100 | >100 | >100 | >100 |
| SK-MEL-2 | >100 | >100 | 55.6 | 91.4 | 26.7 | 22.7 | >100 |
| SK-MEL-28 | >100 | >100 | 49.2 | >100 | 19.9 | >100 | >100 |
| SK-MEL-5 | >100 | 66.2 | 41.4 | >100 | 9.24 | 7.47 | 65.1 |
| UACC-257 | >100 | >100 | >100 | >100 | >100 | >100 | >100 |
| UACC-62 | >100 | 89.7 | >100 | >100 | 6.31 | 9.02 | 37.3 |

**Table 4.** $IC_{50}$ values of potent compounds against HFF1 cell line.

| SN | $IC_{50}$ (µM) |
|---|---|
| 2 | 21 |
| 3 | >100 |
| 4 | 8.6 |
| 5 | 5.9 |
| 7 | 40 |
| dacarbazine | >100 |

observed for compound **7** against SK-MEL-5 and UACC-62 cell lines. 3-Flurophenyl-derived fused-thiazole (**8**) showed potent $LC_{50}$ values against LOX IMVI and MALME-3M cell lines. Based on the $GI_{50}$, TGI and $LC_{50}$ values, compounds **3** and **5** are the most promising agents in the series. Although our compounds were most potent against LOX IMVI cell line, these cells are semi-adherent, which limited our ability to fix these cells. Hence, for the proof of concept studies below, we used adherent cell lines SK-MEL-5 and SK-MEL-28.

Based on the anti-melanoma potency, we can deduce a structure activity relationship (SAR) of the compounds. Aliphatic substituted compounds did not show any significant anti-melanoma activity. *N*-Aryl compounds with a small hydrophobic substituent such as fluoro, methyl and trifluoromethoxy groups showed significant activities. 3-Hydroxy substituted phenyl group is an exception.

To assess the potential use of lead compounds as anti-melanoma agents, we tested their toxicity for the normal human foreskin fibroblast (HFF-1) cell line taking into consideration that MM is usually surrounded by the cells of this kind. The effects of some of the potent compounds on this cell line were very encouraging (table 4). Phenyl-substituted compound (**2**) inhibited the growth of HFF-1 cells with an $IC_{50}$ value of 21 µM, which was much higher than the $GI_{50}$ values against some of the melanoma cell lines. Trifluoromethoxy-substituted compound (**3**) did not show any notable toxicity against the tested healthy cell line, which is very important for the further drug development of this compound. 3-Fluorophenyl derivative (**5**) was somewhat toxic to HFF1 cells. Compound **7** showed good tolerance to healthy cell lines with the $IC_{50}$ value of 40 µM. Like compound **3**, the positive control, dacarbazine, also did not show any notable toxicity against HFF1 cells. $IC_{50}$ values of compounds **1** and **8** were not determined against the HFF-1 cell line for their low solubility in the growth media.

After determining the selective toxicity of potent molecules, we determined their mode of action against the melanoma cell line. Apoptosis or programmed cell death is a physiological process to remove unwanted or damaged mammalian cells. We detected apoptosis mode of cell death by our potent compounds using flow cytometry-based Annexin V assay. Compound **3** caused 23% apoptosis

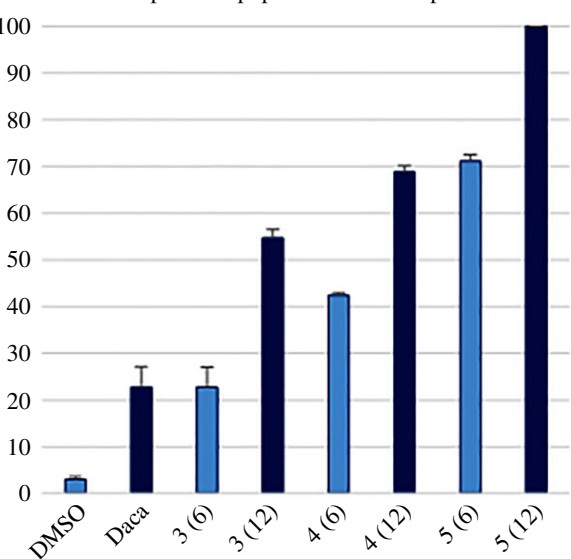

percent apoptitic cells in sample

**Figure 3.** Apoptosis measurements using Annexin V flow cytometry assay of compounds **3** and **5** against SK-MEL-28 cell line at 6 and 12 µM concentrations. DMSO and dacarbazine (Daca) are negative and positive controls, respectively. Mean ± s.e.m., $n = 3$ per group.

at 6 µM and 54% apoptosis at 12 µM (figure 3). 3-Fluorophenyl derivative (**5**) was a very potent apoptosis-inducing agent that caused 100% cell death at 12 µM concentration. Dacarbazine, the positive control, caused 22% apoptotic cell death at 200 µM concentration. This very high concentration effected 22% apoptosis, which shows the high potency of our novel compounds against melanoma cell line.

Apoptosis is usually associated with profound DNA fragmentation that makes the cell death irreversible. DNA fragmentation could be due to the increasing of nuclear DNA access to apoptotic endonucleases and the induction of these enzymes [18,19]. This DNA fragmentation is best measured by using the terminal deoxynucleotidyl transferase dUTP nick-end labelling (TUNEL) assay [19]. Our TUNEL staining for DNA fragmentation and irreversible cell death showed that all three test compounds (**3**, **4** and **5**) were cytotoxic to melanoma cells (figure 4a,c), and the cytotoxicity seemed to be dose-dependent. Surprisingly, the cytotoxicity of dacarbazine was negligent, and at lower concentration, it even seemed to be cytoprotective. In all three compounds as well as in dacarbazine, the cytotoxicity at 10 µM was higher than at 5 µM concentration. Interestingly, the cytotoxicity of the two test compounds was higher than dacarbazine (our positive control). It did not reach statistical significance for compound **3** at 10 µM versus dacarbazine at 10 µM, but it was significant in two other compounds (**4** and **5**) versus dacarbazine at 10 µM. Based on these data, and taking them together with the selectivity towards melanoma versus non-melanoma cells, we concluded that our new compounds have promise as new anti-melanoma drugs. Immunostaining for major marker of oxidative injury, heme oxygenase-1 (HO1) was done to determine whether the test compounds induce oxidative injury. The measurement of mean intensity of colour, which illustrates the level of protein expression, showed it was induced by the exposure with the two out of three compounds (figure 5b,c). Contrary to HO1, no induction of CAD or EndoG was observed (figure 4d,e). Apparently, none of these compounds induced cell death endonucleases. Activation of endonucleases and their translocation to the nucleus is actually the last step when cell death becomes irreversible. It is often preceded by an oxidative cell injury that activates or aggravates the cell death mechanism. To measure whether our compounds induce oxidative injury, we chose to measure HO1 which is a well-established mechanistic marker of oxidative injury [20,21]. Because HO1 is not a nuclear marker and it is not translocated to the nucleus and contact DNA during cell death, instead of colocalization by pixels, we counted the percentage of TUNEL-positive (dead) cells which are also HO1-positive (figure 5a). Similar to the previous experiments, we observed strong elevation of HO1 in TUNEL-positive cells after exposure to the compounds (figure 5b). The elevation was dose-dependent. However, dacarbazine killed melanoma cells without induction of HO1. We next examined whether caspase-activated DNase (CAD), a known mechanistic marker of caspase-mediated apoptosis [22,23],

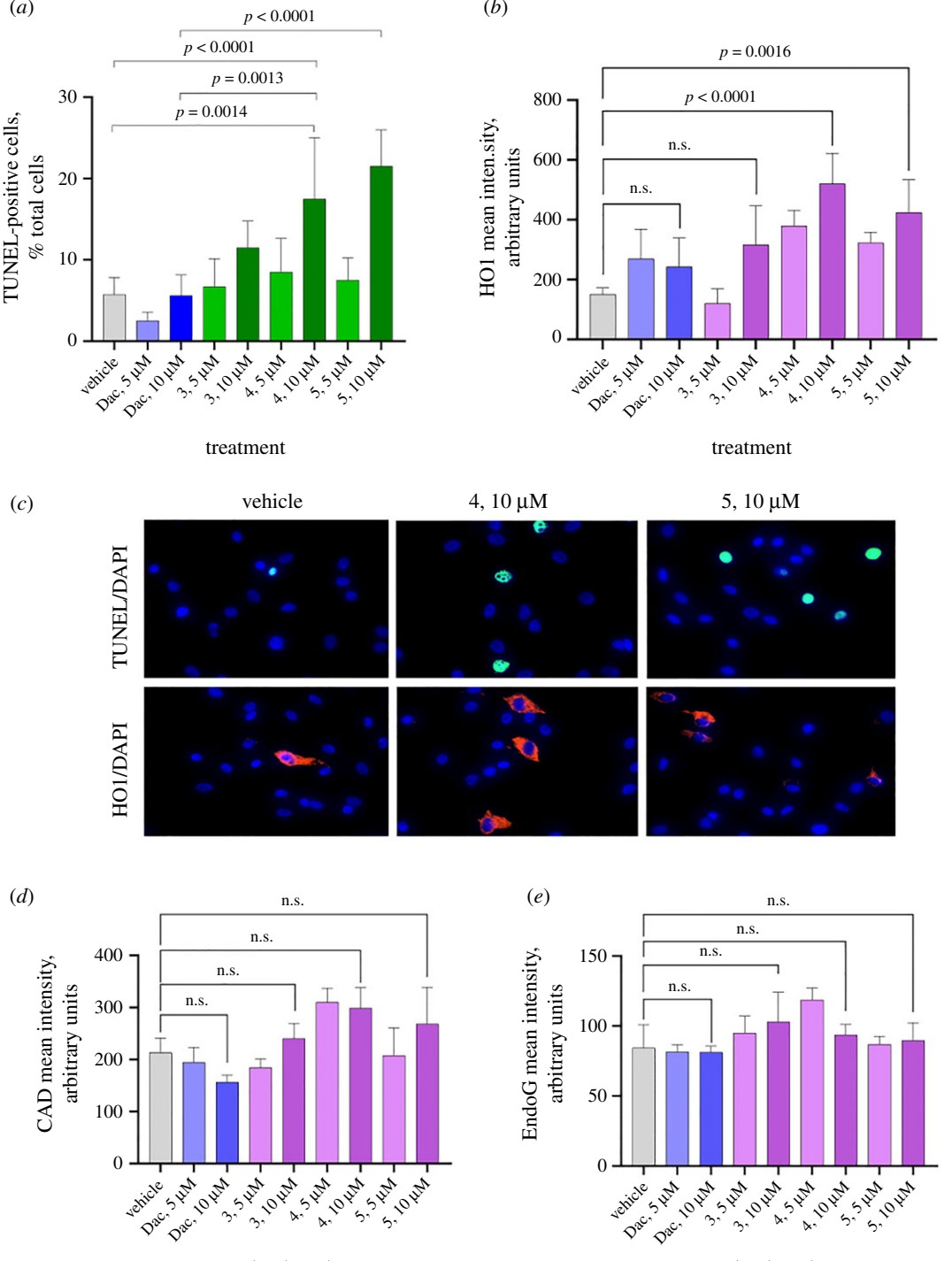

**Figure 4.** TUNEL and immunocytochemistry staining of melanoma cells exposed with vehicle (DMSO), dacarbazine (Dac) and three compounds (**3**, **4** and **5**) under investigation at 5 or 10 μM for 24 h. (*a,b*) TUNEL-positivity and HO1 mean intensity induced by compounds **4** and **5** at 10 μM concentrations. (*c*) Representative images. (*d,e*) No significant induction observed for CAD or EndoG mean intensity. Mean ± s.e.m., *n* = 4 per group.

is colocalized with TUNEL during cell death induced by dacarbazine or our test compounds (**3**, **4** and **5**). These compounds induced the colocalization of CAD with TUNEL indicating that these chemicals were capable of inducing caspase-mediated apoptosis in addition to the caspase-independent one (figure 5*c*). For some reason, no dose dependency was observed, the colocalization was at the same level at 5 and 10 μM. Apparently, the CAD/TUNEL association had an upper limit and reached a plateau. Unlike our compounds, dacarbazine did not induce any association of CAD with TUNEL.

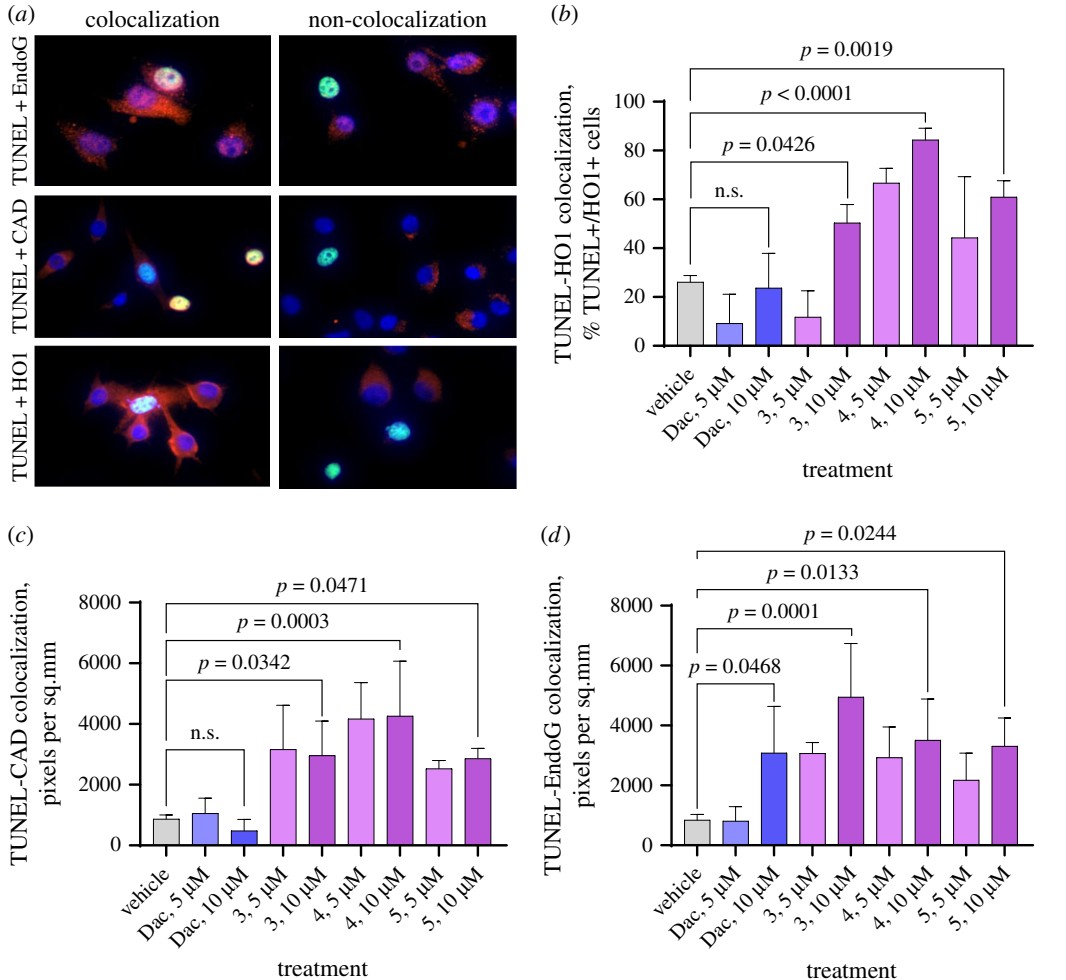

**Figure 5.** TUNEL and immunocytochemistry staining of melanoma cells exposed with vehicle (DMSO), dacarbazine (Dac) and three compounds under investigation at 5 or 10 μM for 24 h. (*a*) Representative images of TUNEL colocalized or not colocalized with each of the three protein markers used in this study. (*b*) All three compounds induced significant oxidative injury as measured by the strong association of HO1 staining in TUNEL-positive cells, while dacarbazine did not induce it. (*c*) All three compounds also induced CAD-associated TUNEL indicative of caspase-mediated apoptosis, whereas dacarbazine did not induce it. (*d*) All four compounds induced TUNEL colocalized with EndoG suggesting that the cell death was mediated by EndoG, a marker of non-caspase-mediated apoptosis. Mean ± s.e.m., $n = 4$ per group.

This suggested dacarbazine did not induce caspase-mediated apoptosis at the used concentrations. Thus, our compounds had two mechanisms while dacarbazine had only one mechanism of cell death in this model. To determine whether there may be an association between DNA fragmentation measured by TUNEL assay, and immunocytochemistry (ICC) of EndoG, the marker of caspase-independent apoptosis [21,24], and the most active endonuclease in cancer cells [25,26] known to translocate to the nucleus during cell death [27]. The colocalization analysis showed that dacarbazine and our compounds at 10 μM induced a significant increase of EndoG overlap with TUNEL staining indicating that the cell death induced by the drug and the new chemicals is likely EndoG-mediated (figure 5*d*). The increase was dose-dependent, and dacarbazine at 5 μM did not show any effect, while our compounds even at 5 μM showed the increase of EndoG-TUNEL association. Overall, the effect of our compounds in terms of EndoG involvement seemed to be similar to that of dacarbazine; all three compounds and dacarbazine induced caspase-independent apoptosis.

Thus, to determine the mechanism of action of our compounds on melanoma cells *in vitro*, we applied the fluorescent TUNEL assay combined with quantitative ICC for three markers: HO1 for oxidative cell injury, CAD for caspase-mediated apoptosis and EndoG for caspase-independent apoptosis. Our TUNEL data showed that compounds are more toxic to melanoma cells than dacarbazine. Two of three compounds (**4** and **5**) induced oxidative injury as measured by the increase mean intensity of HO-1. However, none of the compounds or dacarbazine induced apoptotic endonucleases. This is not

surprising because apoptotic endonucleases are commonly present in the cell in sufficient amounts to be able to destroy DNA upon cell death without induction (which most likely would not be possible in a dead cell).

Because TUNEL assay measures DNA fragmentation by endonucleases, we use ICC for two endonucleases, EndoG and CAD, to determine the mode of cell death. Our experiments showed that our compounds induce both caspase-independent and caspase-dependent apoptosis, as well as oxidative injury to melanoma cells. Contrary to our compounds, the currently used anti-melanoma drug dacarbazine induced only caspase-independent apoptosis. No caspase-dependent apoptosis or oxidative injury was observed after exposure of melanoma cells to the drug. The fact that compounds use three cytotoxic mechanisms instead of one explains why our thiazole compounds are more effective.

In conclusion, we have discovered potent anti-melanoma agents by screening a panel of 50 compounds and eight of these compounds showed potent anti-melanoma properties with $GI_{50}$ values at low single digit µM concentration. Some of these compounds are non-toxic or less toxic to healthy HFF1 cell lines. Mode of action has been determined by flow cytometry and TUNEL assay. Potent molecules were found to be apoptotic and DNA-fragmenting agents. The mode of action studies determined that our new compounds simultaneously use three cytotoxic mechanisms to kill melanoma cells. *In vivo* anti-melanoma and ADMET studies will be reported in due course.

Data accessibility. Data has been provided as supplementary material.

Authors' contributions. S.A.C. carried out *in vitro* cytotoxicity studies; M.N. synthesized the compounds to replenish the stock; M.M.F. determined the cytotoxicity of the compounds; A.V.S. carried out immunocytochemistry; A.G.B. designed the immunocytochemistry protocol, and M.A.A. designed and supervised the *in vitro* experiments and obtained funding.

Competing interests. We declare we have no competing interests.

Funding. This publication was made possible by the Arkansas INBRE program, supported by a grant no. P20 GM103429 from the National Institute of General Medical Sciences (NIGMS), National Institutes of Health (NIH) and the Winthrop P. Rockefeller Cancer Institute at the University of Arkansas for Medical Sciences (UAMS). NIH grant no. P20 GM109005 (AGB); ABI-mini grant no. 200138; and VA Merit Review grant no. 2IO1BX002425 (AGB).

Acknowledgements. We are thankful to the NCI Development Therapeutics Program (DTP) for cytotoxicity data.

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
