## [Peer Review File · Royal Society Open Science]

Review History

RSOS-210395.R0 (Original submission)

Review form: Reviewer 1

Is the manuscript scientifically sound in its present form?

Yes

Are the interpretations and conclusions justified by the results?

Yes

Is the language acceptable?

Yes

Do you have any ethical concerns with this paper?

No

Have you any concerns about statistical analyses in this paper?

No

Recommendation?

Accept with minor revision (please list in comments)

Comments to the Author(s)

The present manuscript describes biological evaluation as antimelanoma agents of a series of steroidal thiazole derivatives previously prepared in the Alam group. The manuscript presents interesting results, and provides much information regarding the investigations on the mechanism of action of the thiazole derivatives. Therefore, I would like to recommend this manuscript for publication in Royal Society Open Science, after the following comments are addressed:

- The authors synthesized and submitted 50 different thiazole derivatives to the NCI, and from there, compounds 1-8 were selected for further testing. Please include what criteria was used to include compounds 1-8 and exclude the rest? Was this based on a given GI50 cutoff?
- A follow-up question arises upon examination of the data in table 1, as compound 6 clearly presents a much narrower activity profile. Why was compound 6 selected for further testing in the first place?
- The authors could provide a more concise medicinal chemistry analysis, for example choosing a representative melanoma cell line, they could provide information on any structure-activity trends. Do more lipophilic or more polar aromatic rings increase activity? If no clear relationships are identified, please state it.
- Which compounds are taken to next stage looking at ADME and in-vivo antimelanoma? Why? Is compound 6 still being considered?

Review form: Reviewer 2

Is the manuscript scientifically sound in its present form?

Yes

Are the interpretations and conclusions justified by the results?

Yes

Is the language acceptable?

Yes

Do you have any ethical concerns with this paper?

No

Have you any concerns about statistical analyses in this paper?

No

Recommendation?

Major revision is needed (please make suggestions in comments)

Comments to the Author(s)

This is an interesting work. This manuscript reports the GI50, TGI and LC50 of some thiazolo-androstenone derivatives on melanoma cell lines, for studying the structure-activity relationship. The authors proved that some of these compounds can induce caspase-independent apoptosis, and compounds (3 and 5) strongly induced oxidative injury to melanoma cells as measured by TUNEL colocalization with heme oxygenase-1 (HO1). The authors claim that they have found effective anti-melanoma drugs.

However, there are still some concerns here.

1. On page 4, line 31, the author says that "Overall, amino-thiazole derivatives (Table 1, 1-7)15 were more toxic to melanoma cell lines than phenyl thiazole derivatives." However, no compound has a lower GI50 than compound 8 in all melanoma cell lines. And more importantly, there is no intuitive data to explain that amino-thiazole derivatives (Table 1, 1-7)15 were more toxic to melanoma cell lines than phenyl thiazole derivatives.
2. On page 4, line 47, the author says that " All eight compounds inhibited this cell line with GI50 values as low as 1.64 μM ", however, if the author can use the highest GI50 3.22 μM , the article might be more reliable.
3. On page 6, line 19, the author says that " Phenyl-substituted compound (2) inhibited the growth of HFF-1 cells with an IC50 value of 21 μM , which was much higher than the GI50 values against the melanoma cell lines." However, the IC50 value of compound (2) inhibiting HFF-1 cells is only significantly different from the GI50 value of melanoma cell lines LOX IMVI and MDA-MB-435.

Decision letter (RSOS-210395.R0)

Dear Dr Alam:

Title: Antimelanoma Activities of Chimeric Thiazole-Androstenone Derivatives
Manuscript ID: RSOS-210395

The editor assigned to your manuscript has now received comments from reviewers. We would like you to revise your paper in accordance with the referee and Subject Editor suggestions which can be found below (not including confidential reports to the Editor). Please note this decision does not guarantee eventual acceptance.

Please submit your revised paper before 04-Jun-2021. Please note that the revision deadline will expire at 00.00am on this date. If we do not hear from you within this time then it will be assumed that the paper has been withdrawn. In exceptional circumstances, extensions may be possible if agreed with the Editorial Office in advance. We do not allow multiple rounds of revision so we urge you to make every effort to fully address all of the comments at this stage. If deemed necessary by the Editors, your manuscript will be sent back to one or more of the original reviewers for assessment. If the original reviewers are not available we may invite new reviewers.

When submitting your revised manuscript, you must respond to the comments made by the referees and upload a file "Response to Referees" in "Section 6 - File Upload". Please use this to document how you have responded to the comments, and the adjustments you have made. In

order to expedite the processing of the revised manuscript, please be as specific as possible in your response.

On behalf of the Subject Editor Professor Anthony Stace and the Associate Editor Dr Andrew Harned.

RSC Associate Editor:

Comments to the Author:

The referees have shown some enthusiasm for this work and I believe it is appropriate for this journal. However, they have raised several reasonable queries that would strengthen the manuscript and clarify the author's message. The authors should carefully consider these comments before submitting a revised manuscript.

RSC Associate Editor: 2

Comments to the Author:

The reviewers have raised several valid questions during this review. Addressing these questions in a revised manuscript would strengthen the paper and would be helpful to future readers.

RSC Subject Editor:

Comments to the Author:

(There are no comments.)

Reviewers' Comments to Author:

Reviewer: 1

Comments to the Author(s)

The present manuscript describes biological evaluation as antimelanoma agents of a series of steroidal thiazole derivatives previously prepared in the Alam group. The manuscript presents interesting results, and provides much information regarding the investigations on the mechanism of action of the thiazole derivatives. Therefore, I would like to recommend this manuscript for publication in Royal Society Open Science, after the following comments are addressed:

- The authors synthesized and submitted 50 different thiazole derivatives to the NCI, and from there, compounds 1-8 were selected for further testing. Please include what criteria was used to include compounds 1-8 and exclude the rest? Was this based on a given GI50 cutoff?

- A follow-up question arises upon examination of the data in table 1, as compound 6 clearly presents a much narrower activity profile. Why was compound 6 selected for further testing in the first place?
- The authors could provide a more concise medicinal chemistry analysis, for example choosing a representative melanoma cell line, they could provide information on any structure-activity trends. Do more lipophilic or more polar aromatic rings increase activity? If no clear relationships are identified, please state it.
- Which compounds are taken to next stage looking at ADME and in-vivo antimelanoma? Why? Is compound 6 still being considered?

Reviewer: 2

Comments to the Author(s)

This is an interesting work. This manuscript reports the GI50, TGI and LC50 of some thiazolo-androstenone derivatives on melanoma cell lines, for studying the structure-activity relationship. The authors proved that some of these compounds can induce caspase-independent apoptosis, and compounds (3 and 5) strongly induced oxidative injury to melanoma cells as measured by TUNEL colocalization with heme oxygenase-1 (HO1). The authors claim that they have found effective anti-melanoma drugs.

However, there are still some concerns here.

1. On page 4, line 31, the author says that "Overall, amino-thiazole derivatives (Table 1, 1-7)15 were more toxic to melanoma cell lines than phenyl thiazole derivatives." However, no compound has a lower GI50 than compound 8 in all melanoma cell lines. And more importantly, there is no intuitive data to explain that amino-thiazole derivatives (Table 1, 1-7)15 were more toxic to melanoma cell lines than phenyl thiazole derivatives.
2. On page 4, line 47, the author says that " All eight compounds inhibited this cell line with GI50 values as low as 1.64 μM ", however, if the author can use the highest GI50 3.22 μM , the article might be more reliable.
3. On page 6, line 19, the author says that " Phenyl-substituted compound (2) inhibited the growth of HFF-1 cells with an IC50 value of 21 μM , which was much higher than the GI50 values against the melanoma cell lines." However, the IC50 value of compound (2) inhibiting HFF-1 cells is only significantly different from the GI50 value of melanoma cell lines LOX IMVI and MDA-MB-435.

Author's Response to Decision Letter for (RSOS-210395.R0)

See Appendix A.

Decision letter (RSOS-210395.R1)

Dear Dr Alam:

Title: Antimelanoma Activities of Chimeric Thiazole-Androstenone Derivatives
Manuscript ID: RSOS-210395.R1

It is a pleasure to accept your manuscript in its current form for publication in Royal Society Open Science. The chemistry content of Royal Society Open Science is published in collaboration with the Royal Society of Chemistry.

On behalf of the Subject Editor Professor Anthony Stace and the Associate Editor Dr Andrew Harned.

RSC Associate Editor
Comments to the Author:
(There are no comments.)

Reviewer(s)' Comments to Author:

Appendix A

Reviewers' Comments to Author:

Reviewer: 1

Comments to the Author(s)

The present manuscript describes biological evaluation as antimelanoma agents of a series of steroidal thiazole derivatives previously prepared in the Alam group. The manuscript presents interesting results, and provides much information regarding the investigations on the mechanism of action of the thiazole derivatives. Therefore, I would like to recommend this manuscript for publication in Royal Society Open Science, after the following comments are addressed:

- The authors synthesized and submitted 50 different thiazole derivatives to the NCI, and from there, compounds 1-8 were selected for further testing. Please include what criteria was used to include compounds 1-8 and exclude the rest? Was this based on a given GI50 cutoff?
 - *We appreciate the reviewer for this insightful comment. The cutoff is based on the activity of the compounds against NCI-60 cancer cell lines at 10 μ M. Compounds showing significant activity were tested at lower concentration to determine the IC50 values and these 8 compounds are in the cutoff list.*
- A follow-up question arises upon examination of the data in table 1, as compound 6 clearly presents a much narrower activity profile. Why was compound 6 selected for further testing in the first place?
 - *We want to emphasize that we focused on compounds 3, 5, and 7 in the manuscript.*
- The authors could provide a more concise medicinal chemistry analysis, for example choosing a representative melanoma cell line, they could provide information on any structure-activity trends. Do more lipophilic or more polar aromatic rings increase activity? If no clear relationships are identified, please state it.
 - *We thank the reviewer for this comment. We added the test for more structure-activity trends.*
- Which compounds are taken to next stage looking at ADME and in-vivo antimelanoma? Why? Is compound 6 still being considered?
 - *For the future studies, we are making more derivatives to find better compounds. Due to high selectivity, we are focusing on compounds 3 and 7 for ADMET and in vivo studies.*

Reviewer: 2

Comments to the Author(s)

This is an interesting work. This manuscript reports the GI50, TGI and LC50 of some thiazolo-androstenone derivatives on melanoma cell lines, for studying the structure-activity relationship. The authors proved that some of these compounds can induce caspase-independent apoptosis, and compounds (3 and 5) strongly induced oxidative injury to melanoma cells as measured by TUNEL colocalization with heme oxygenase-1 (HO1). The authors claim that they have found effective anti-melanoma drugs.

However, there are still some concerns here.

1. On page 4, line 31, the author says that "Overall, amino-thiazole derivatives (Table 1, 1-7)15 were more toxic to melanoma cell lines than phenyl thiazole derivatives." However, no compound has a lower IG50 than compound 8 in all melanoma cell lines. And more importantly, there is no intuitive data to explain that amino-thiazole derivatives (Table 1, 1-7)15 were more toxic to melanoma cell lines than phenyl thiazole derivatives.

- *We appreciate the reviewer for this insightful comments. This is a general statement for the series of 50 compounds. Only one of the ~20 phenyl derivative showed potent activity. In the amino series, we found 7 compounds in a series of ~30 compounds as potent antimelanoma agents.*

2. On page 4, line 47, the author says that " All eight compounds inhibited this cell line with GI50 values as low as 1.64 μM ", however, if the author can use the highest IG50 3.22 μM , the article might be more reliable.

➤ *We changes the statement accordingly. We thank the reviewer for this comment to improve the article.*

3. On page 6, line 19, the author says that " Phenyl-substituted compound (2) inhibited the growth of HFF-1 cells with an IC50 value of 21 μM , which was much higher than the GI50 values against the melanoma cell lines." However, the IC50 value of compound (2) inhibiting HFF-1 cells is only significantly different from the GI50 value of melanoma cell lines LOX IMVI and MDA-MB-435.

➤ *Thanks for this useful comment; we changed the statement in the manuscript.*